# The anthropometric assessment of body composition and nutritional status in children aged 2–15 years: A cross-sectional study from three districts in Bangladesh

**Md. Kamruzzaman**  **\*, Shah Arafat Rahman, Sharmin Akter, Humaria Shushmita, Md. Yunus Ali, Md Adnan Billah, Md. Sadat Kamal, M. Toufiq Elahi, Dipak Kumar Paul**

Department of Applied Nutrition and Food Technology, Islamic University, Kushtia, Bangladesh

\* mkzaman.m@gmail.com, md.kamruzzaman@anft.iu.ac.bd

## Abstract

### Background

Early life nutrition plays a critical role in the development of better health and nutrition in adulthood. However, assessing the nutritional status of Bangladeshi children and adolescents through measurement of body composition using skinfold thickness is barely studied. The current study aims to determine children's body composition and nutritional status, and contributing factors among children aged 2 to 15 years in the northern part of Bangladesh.

### Methods

This is a descriptive cross-sectional study done in Bangladesh. Anthropometric methods, including multiple skinfold thickness and basic anthropometric and socio-demographic characteristics, were used. Body composition was calculated from multiple skinfold thicknesses using the standard regression equation. Nutritional status was measured using Z score according to WHO 2007 reference standard. A total of 330 children from Naogaon, Bogra and Kurigram districts in Bangladesh were examined from April 2019 to September 2019.

### Results

The Nutritional status of 2–15 years old child is exceedingly poor in the northern part of Bangladesh. Fat mass and fat-free mass were higher among children from Kurigram district than from Bogra and Naogaon district. Body fat percentages and arm fat area were greater among female children than males. The overall prevalence of stunting, underweight and wasting was around 25%, 32% and 29%, respectively, and the rate was higher among girls and children aged 2–5 years. The average SD score for weight-for-age, height-for-age, and BMI-for-age was -1.295, -0.937 and -1.009. The median weight-for-age and height-for-age Z scores of boys and girls were below the WHO reference percentile rank. Girls were twice (OR:1.951, CI:1.150–3.331) as likely to suffer from being underweight than boys. Children who don't practice handwashing are three times (OR:3.531, CI:1.657–7.525) more likely to

**Data Availability Statement:** All relevant data are within the manuscript and its Supporting Information files.

**Funding:** The authors received no specific funding for this work.

**Competing interests:** The authors have declared that no competing interests exist.

be underweight. Children become underweight and stunted when their family income is not sufficient to maintain their nutritional requirements.

## Conclusions

The children of the three northern districts had a poor nutritional status, and family income was the potential contributing factor. Therefore, interventions like the promotion of income-generating activities and integrated approaches to ensuring food diversification could be an option to address the nutritional problem of children of the three northern districts of Bangladesh.

## Introduction

Malnutrition is a term that may seem to general people as undernutrition; however, theoretically, both under-nutrition and over-nutrition are referred to as malnutrition [1]. The coexistence of both undernutrition and overnutrition is now known as the double burden of malnutrition. When the essential nutrient intake through diet does not meet the maintenance and growth and development, it is termed undernutrition. Whereas excessive intake, compared to requirements of nutrients, is termed over-nutrition [2]. All over the world, people are confronted with a rising prevalence of overweight and obesity, along with the existing high prevalence of undernutrition [3]. The impact of undernutrition or overnutrition during the early stage of life may persist throughout life. Morbidity and mortality are the primary consequences of malnutrition among children worldwide [4] and act as a vicious cycle through the life cycle [5]. The common form of malnutrition among children is stunting, wasting and underweight, and overweight and obesity. Over the last couple of decades, the prevalence of malnutrition decreased, though not satisfactorily. Worldwide 5.2 millions under-5 children died annually [6], and 3.5 millions of this death and 35% of morbidities are caused by malnutrition either directly or indirectly [7], with most living in developing countries [8].

Most importantly, 68% of the world's wasted and 55% of stunted children are reported to be in Asia, while 14.6% wasted and 32.7% stunted children in South Asia [9]. Bangladesh has celebrated a reportable decline in the rate of malnutrition prevalence over the last few decades; however, the current rate is not within the acceptable range. According to the BDHS report, the stunting rate has declined from 61% in 1995 to 31% in 2017, while wasted and underweight prevalence has declined from 21% to 8% and from 52% to 22%, respectively [10]. Rates of malnutrition in Bangladesh are among the highest globally, and more than 54% of preschool-age children, equivalent to over 9.5 million children, are malnourished [11]. The prevalence of malnutrition is alarmingly higher among female adolescents in Bangladesh [12], and the rate is still high to achieve the SDGs aim to end all forms of malnutrition by 2030 [13].

Infield studies, anthropometric measurements, like Body Mass Index (BMI), Fat Mass Index (FMI), Fat-Free Mass Index (FFMI), waist circumference, waist-hip ratio, and multiple skinfold thickness, are used to predict fat mass and fat-free mass. Though these techniques lack precision, they hold a few advantages in the field survey over others, being less expensive, easy to carry, noninvasive. The gold standard for measuring body composition has been DXA, where low dose X-ray is used for whole-body measurements of adipose tissue or fat and lean tissue [14, 15]; however, they may not be suitable for field surveys. In children, there has been a resurgence of interest in body composition [16]. The proliferation of new measurement techniques would be measuring the subcutaneous fat layer, namely skinfold thickness. Skinfold

thickness measurements are said to provide an estimate of the size of the subcutaneous fat depot, which, in turn, provides an estimate of the total body fat. Variations in the distribution of subcutaneous fat occur with sex, race, and age. A combination of skinfold measurements can estimate body density, which can derive body fat percentage using an empirical equation. The skinfold measurement is a well-established means of assessing the subcutaneous fat at all ages, including infancy and neonatal period. The measurement is relatively easy, fast, non-invasive, and requires simple equipment. Body composition measurement using skinfold thickness involves few limitations like within-examiner error, between-examiner error, and another limitation: total body fat cannot be obtained from one skinfold site measurement [17]. Thus, this method requires multiple skinfold measurements, and researchers cannot depend on one site measurement. Brewis (2011) stated that skinfold thickness measurements are standard practice assessing nutritional status by bio-cultural anthropologists, nutritional anthropologists, and human biologists who engage in fieldwork. Anthropometric indicators are combined to form anthropometric indices, measuring children's body composition and nutritional status [18].

Malnutrition during childhood is a critical issue and depends on multiple complex and interrelated issues. Studies on children's body composition and nutritional status in Bangladesh are challenging because of the country's large population size, higher rate of poverty and illiteracy, socioeconomic disparities, and backwardness. In addition, people from the northern part of Bangladesh are susceptible to natural disasters like floods, drought, and poverty. Thus, studies to assess the nutrition status of children from the northern part of Bangladesh would add value to the policymakers to make a link with socioeconomic variables and address those issues properly. However, studies on the nutritional status of Bangladeshi children are primarily based on anthropometric measurements of height and weight, and body composition measurements using skinfold thickness are limited. The present study aims to investigate body composition and nutritional status of children aged 2–15 years, incorporating anthropometric indicators in three different districts, and comparing them with a reference standard. Moreover, to determine the impact of the socio-economic factors that may affect malnutrition.

## Subjects and methods

### Study design and subject

This study was conducted in the three northern districts, Bogra, Naogaon, and Kurigram, Bangladesh (Fig 1). The participants were selected utilizing of multiple steps, simple random sampling, considering first the location (Bogra, Naogaon, and Kurigram district) and then the random assignment of the villages within each district. It was a descriptive cross-sectional survey, and a total of 330 children (age 2–15 years) were selected for the purpose. The sample size was calculated using the single population proportion formula by considering the following assumptions: Proportion = 30% (proportion of malnutrition among children), margin error = 5%, CI = 95%, and the sample size obtained was 323 [19]. A detailed questionnaire was used to collect data, including parent education level, parent occupation, age, family history, and family income. Age, level of education, family income was collected from direct questionnaires. A trained survey team consisting of three members was assigned to visit the households and conduct the survey. A repeated pretest ensured the quality of the interview and examination. At the end of each day, the experts evaluated the collected raw data to ensure completeness and consistency. If any inconsistency or incompleteness were noticed, households were revisited to remove those inconsistencies. The study was conducted following the 1964 Declaration of Helsinki and its later amendments. Informed parental written consent or consent from parents or guardians or caregivers and the child's assent regarding the nature and

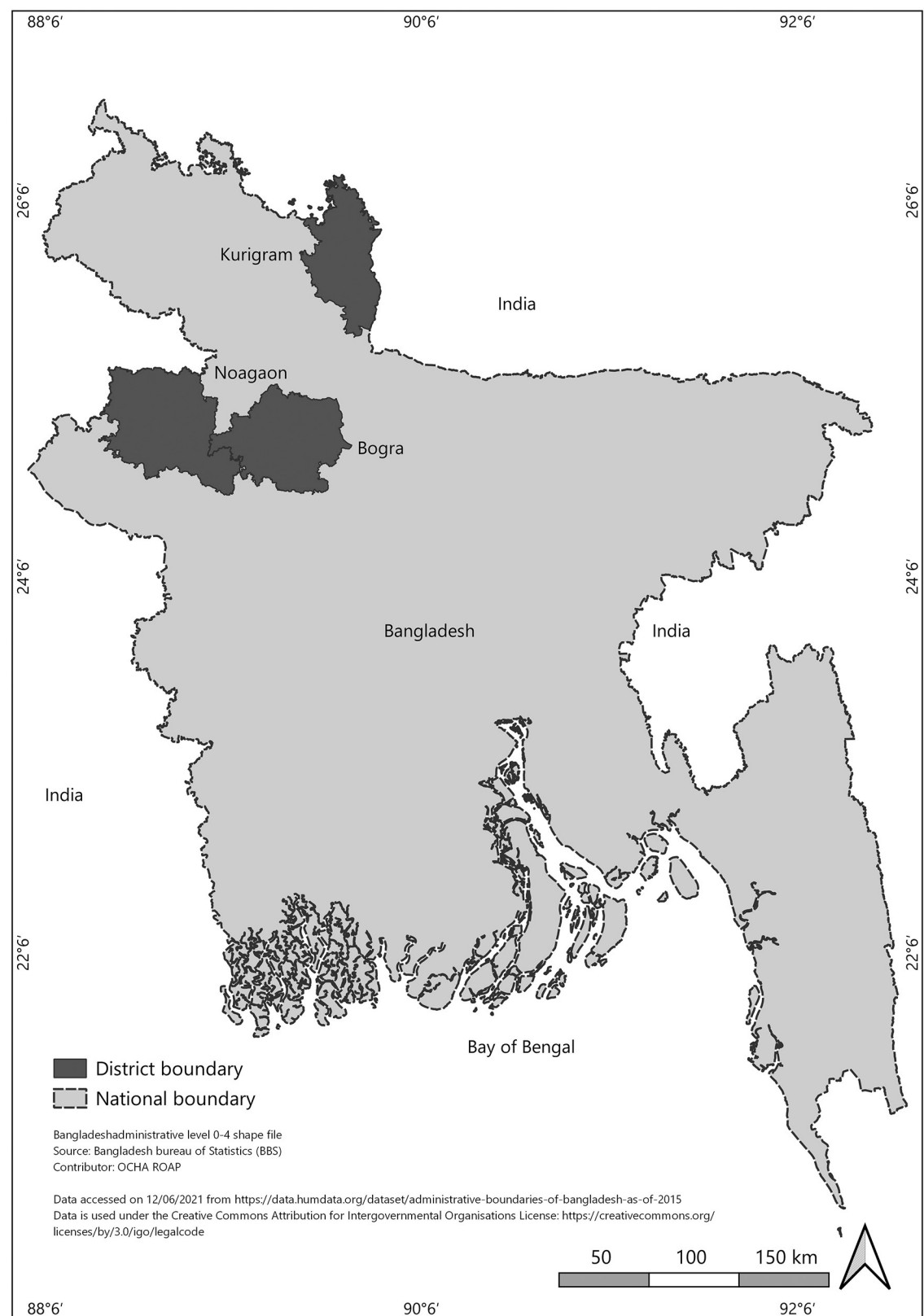

**Fig 1. Study area of three northern districts (Bogra, Naogaon and Kurigram) of Bangladesh [reprinted from the humanitarian data exchange [63] under a creative common attribution 4.0 international license].**

purpose of the study were obtained. It was also confirmed the personal information would be kept confidential. Mothers or caregivers with children ≥2 years and ≤15 years willing to participate in the study were only selected. Moreover, children with malnutrition not resulting from insufficient dietary intake, for example, cystic fibrosis, metabolic and endocrine disorders, or other disorders, were excluded [16]. A detailed questionnaire containing both closed and open-ended questions was used to collect the socio-economic characteristics. The Ethical Review Committee approved the study and study protocol (Ethical approval no: FBS/ERC/2019), Faculty of Biological Science, Islamic University, Kushtia, Bangladesh. The study was conducted under the Dept. of Applied Nutrition & Food Technology, Islamic University, Kushtia, Bangladesh.

## Anthropometric measurement

All anthropometric measurements were performed using standardized protocol, instruments, and conditions [20, 21]. Trained technicians performed the measurements under standardized conditions. Before measuring, all the equipment was calibrated and validated. The mother and/or caregivers of the children were asked for a face-to-face interview. All measurements were taken three times, and an average value was recorded. Weight was measured to the nearest 0.1 kg using an electronic scale (Omron HBF-375 Karada Scan, Japan). During measurement of weight, minimum clothing and bare feet were ensured. Height was measured at the nearest 0.1 cm using a static digital height measurer (Omron HBF-375 Karada Scan, Japan). MUAC was measured at the midpoint of the upper left arm after locating the middle; the left arm is extended to hang loosely by the side, with the palm facing inward. The tape is wrapped gently but firmly around the arm at the midpoint, care being taken to ensure that the arm is not squeezed. Measurements were taken to the nearest millimeter using a standard MUAC tape (Ibis Medical, Kerala, India). Skinfold thickness was measured to the nearest 0.1 mm using a Harpenden skinfold calliper (Chasmors Ltd, London, UK). Skinfold thickness from four anatomical sites (Triceps, Biceps, Subscapular, and Suprailiac) of the right side of the body were measured. The guidelines for anatomy landmarks recommended by the International Society for the Advancement of Kinanthropometry were followed [22]. Triplicate measurements were performed at all points, and an arithmetic average for each of the anatomical points was taken. Z-scores for weight for height (WFH), weight-for-age (WFA), height-for-age (HFA), and body mass index (BMI)-for-age (BFA) were calculated using growth standards following WHO reference standard. A cut-off of <-2 and <-3 Z scores for Weight-for-age, height-for-age, and Weight-for-height/BMI-for-age was considered moderate and severe underweight, stunted, and wasted, respectively, whereas a cutoff of >-2 and >-3 Z score for Weight-for-height/BMI-for-age was considered as overweight and obesity, respectively [23, 24]. The MUAC cut-off points of >135 mm for children aged 2–5 years is deemed to be expected, and a MUAC <115 mm is considered as severe acute malnutrition (SAM) [25, 26].

**Body composition measurement.** A two-compartment model was used to measure the fat mass and fat-free mass. First, the predictive value of body fat percentage was calculated using a published equation. Then, triceps, biceps, subscapular, and suprailiac skinfold thickness was used in the equation developed by Bary et al. (2001) (Eq 1) [27]. First fat mass (FM) was calculated from percent body fat, and using the theory of two-compartment body composition model, fat-free mass (FFM) was calculated by subtracting the fat mass (FM) from total body weight [28, 29]. Next, MUAC and triceps skinfold thickness were used to measure the arm muscle area and arm fat area using the standard equation [16, 30] *[Bary et al. (2001)]*.

$$\begin{aligned}BF\ (\%)[Boys\ \&\ Girls]\\= 8.71 + 0.19 \times (Subscapular\ (mm)) + 0.76 \times (Biceps\ (mm) + 0.18\\\times (Suprailiac\ (mm)) + 0.33 \times (Triceps\ (mm))\end{aligned} \qquad (1)$$

Body mass index (BMI) was calculated using the standard equation of WHO (1995) to measure the body composition characteristics of the children [31]. Fat mass index (FMI) and fat-free mass index (FFMI) are similar indexes of body mass index and calculated using the following equation (Eq 2, Eq 3) [29]. It is worth mentioning that mathematically BMI is the summation of FMI and FFMI [32].

$$FMI \left( \frac{kg}{m^2} \right) = \frac{\text{Fat Mass (kg)}}{Height^2(m^2)} \qquad (2)$$

$$FFMI \left( \frac{kg}{m^2} \right) = \frac{\text{Fat Free Mass (kg)}}{Height^2(m^2)} \qquad (3)$$

## Statistical analysis

After collection, data were checked thoroughly again for consistency and completeness. All analysis was done by appropriate statistical methods using RStudio (Version 1.3.1093) based on R (Version 4.0.3). The R package "zscorer" and "anthro" were used to measure SD/Z score using WHO growth standard for children aged 0–60 months and 60–228 months. According to socio-economic characteristics, the relative distribution of the children, adolescent boys, and girls was analyzed using descriptive statistics and was expressed in both numbers and percentages. The results are expressed as mean±standard deviation (x±sd). The normality of the data was tested using Q-Q-plot and Shapiro-Wilk test, and the Levene test was used to test homogeneity of variance. An independent sample t-test was used to measure the differential between two groups, and one-way ANOVA was used for more than two groups. Post hoc analysis (Tukey HSD) was used for determining the mean differences within the groups. Kruskal-Wallis rank-sum test, a nonparametric alternative of one-way ANOVA, was used when the assumption of equal variances and normality assumption was violated. WHO 2007 reference standard was used to measure stunting, wasting, and BMI-for-age Z score. Bivariate analysis was performed to find out the variables which are correlated with body fat mass. Binary and polynomial logistic regression was performed to measure the association of malnutrition with sociodemographic variables. A p-value of $\leq 0.05$ was considered statistically significant.

## Results

### Anthropometric and body composition characteristics

Anthropometric and body composition characteristics are shown in Tables 1–3. The mean MUAC, percent body fat, fat-free mass, and arm muscle area were 162.57±15.38 (mm), 13.56 ±1.26 (%), 22.56±8.04 (kg), and 1801.4±341.30 (mm$^2$) respectively (Table 1). Categorization was done according to three stratified variables like gender, age category, and area of residence (districts). Table 1 compares the anthropometric characteristics between boys and girls. Among the total 330 studied children, more than three-fifth (63%) of the children were boys, and less than two-fifth were girls. All the anthropometric variables, except MUAC, FFM/FM and arm muscle area and family income, were significantly different between boys and girls (p≤0.001). Three among the four studied skin-fold thickness, and their sum was considerably higher among girls than boys (p≤0.05). The skinfold thickness on the biceps was found similar among boys and girls. The average arm muscle area (AMA) was found same among boys and girls (p>0.05); however, arm fat area (AFA) was significantly higher among girls (365.15mm$^2$) compared with boys (333.28mm$^2$). In contrast, boys' average Body Mass Index (BMI) was found a bit higher than girls. The mean BMI of 208 boys was found 15.35, whereas this was 14.26 for 122 studied girls.

**Table 1. Anthropometric and body composition characteristics of children stratified by gender (n = 330).**

| Characteristic | Boys (n = 208) | Girls(n = 122) | Total (n = 330) |
|---|---|---|---|
| | mean (SD) | mean (SD) | mean (SD) |
| Age (Months)** | 91.31 (30.54) | 80.46 (28.61) | 87.3 (30.26) |
| Family Income (BDT)** | 9149.52 (8556.87) | 9570.49 (6998.73) | 9305.1 (8007) |
| Weight (kg)** | 22.80 (8.77) | 18.63 (8.36) | 21.26 (8.84) |
| Height (cm)** | 120.22 (17.81) | 112.83 (15.72) | 117.49 (17.41) |
| MUAC (mm) | 160.70 (17.27) | 160.39 (19.48) | 160.18 (18.09) |
| *Skin-fold Thickness (mm)* | | | |
| Triceps * | 4.32 (1.41) | 4.69 (1.47) | 4.46 (1.44) |
| Biceps | 3.10 (1.08) | 4.44 (16.52) | 3.04 (1.09) |
| Suprailiac** | 2.38 (0.78) | 2.78 (1.12) | 2.53 (0.94) |
| Subscapular* | 3.44 (1.25) | 3.88 (1.81) | 3.61 (1.49) |
| *Sum of Skinfold* | 13.24 (3.41) | 14.28 (4.62) | 13.63 (3.93) |
| *Thickness (mm)** | | | |
| BMI (kg/m$^2$)** | 15.35(3.08) | 14.26(3.64) | 14.95 (3.34) |
| FFMI (Kg/m$^2$)** | 13.26 (2.68) | 12.29 (3.12) | 12.9 (2.88) |
| FMI (kg/m$^2$)** | 2.08 (0.46) | 1.97(0.59) | 2.04 (0.52) |
| Fat Mass (kg)** | 3.11 (1.28) | 2.59 (1.38) | 2.92 (1.34) |
| Body Fat (%) | 13.58 (1.28) | 13.73 (1.56) | 13.63 (1.39) |
| Fat Free Mass (kg)** | 19.69 (7.54) | 16.04 (7.04) | 18.34 (7.56) |
| FFM/FM Ratio | 6.43 (0.66) | 6.37 (0.73) | 6.4 (0.68) |
| Arm Muscle Area (mm$^2$) | 1745.4 (372.78) | 1711.9 (417.15) | 1733.00 (389.48) |
| Arm Fat Area (mm$^2$)* | 333.28 (119.98) | 365.15 (155.75) | 345.06 (134.41) |

N.B.

*p$\leq$0.05

**p$\leq$0.001, BMI = Body Mass Index, FFMI = Fat Free Mass Index

FMI = Fat Mass Index, MUAC = Mid Upper Arm Circumference

Comparison of anthropometric and body composition according to study area are shown in Table 2. Around 90% of children were from two study districts, Bogra and Kurigram and an almost equal number of children from each of these two districts. Age, weight, height, BMI, fat mass, fat-free mass, FMI, FFMI were significantly higher among children of Kurigram district compared to children from Bogra and Naogaon district (Table 2) (p<0.05). The sum of skinfold thickness at four sites of children from Bogra was significantly higher than children from Kurigram. In contrast, this value was similar for children from Naogaon and Bogra (p>0.05) and Naogaon and kurigram (p>0.05). Arm muscle area and arm fat area also follow this pattern of difference. However, when comparing skinfold thickness at a single site, biceps, subscapular, and suprailliac of children from Bogra and Naogaon were significantly different from children from Kurigram. Skinfold thickness at triceps follows a similar pattern of difference as the sum at four sites followed. MUAC and the ratio of fat-free mass to fat mass were similar for children from the three-study area. The average family income of children from Kurigram was found to be significantly lower than the family income of children from Bogra and Naogaon.

We categorized our sample child into five age groups shown in Table 3. The highest proportion (42%) of children were within the age group of 5–8 years, followed by 80–10 years (29%) and 2–5 years (22%). It is clear from Table 3 that the average weight of children of the five-age category was significantly different from each other, except only between-group four and five.

**Table 2. Dwelling-specific distributions and comparison of children characteristics (n = 330).**

| Characteristic | Bogra = 149 | Nagaon = 29 | Kurigram (n = 152) |
|---|---|---|---|
| | mean (SD) | mean (SD) | Mean (SD) |
| Age (Months) | 78.19 (26.44)[a] | 80.49 (30.74)[a] | 97.53 (30.62)[b] |
| Family Income (BDT) | 9474.49 (6926.11)[a] | 11793.10 (6470.42)[a] | 8664.47 (9127.10)[b] |
| Weight (kg) | 17.023 (5.431)[a] | 17.482 (6.537)[a] | 26.12 (9.44)[b] |
| Height (cm) | 112.35 (15.17)[a] | 113.220 (19.438)[a] | 123.34 (17.34)[b] |
| MUAC (mm) | 158.148 (19.493) | 162.724 (22.459) | 162.57 (15.38) |
| *Skin-fold Thickness (mm)* | | | |
| Triceps | 4.804 (1.518)[a] | 4.687 (1.391)[ac] | 4.07 (1.28)[bc] |
| Biceps | 2.82 (1.07)[a] | 2.643 (0.791)[a] | 3.33 (1.09)[b] |
| Subscapular | 3.978 (1.542)[a] | 4.637 (2.373)[a] | 3.04 (0.90)[b] |
| Suprailiac | 2.770 (1.013)[a] | 2.967 (1.544)[a] | 2.21 (0.53)[b] |
| *Sum of Skin-fold Thickness (mm)* | 14.38 (4.31)[a] | 14.94 (5.06)[ac] | 12.64 (2.97)[bc] |
| BMI (kg/m$^2$) | 13.27 (2.20)[a] | 13.30 (2.18)[a] | 16.90 (3.39)[b] |
| Fat Mass (kg) | 2.34 (0.90)[a] | 2.43 (1.06)[a] | 3.57 (1.45)[b] |
| Percent Body Fat | 13.69 (1.51) | 13.68 (1.42) | 13.56 (1.26) |
| Fat Free Mass (kg) | 14.68 (4.62)[a] | 15.05 (5.52)[a] | 22.56 (8.04)[b] |
| FFMI (Kg/m$^2$) | 11.45 (1.83)[a] | 11.48 (1.92)[a] | 14.61 (2.95)[b] |
| FMI (kg/m$^2$) | 1.83 (0.44)[a] | 1.82 (0.34)[a] | 2.29 (0.50)[b] |
| FFM/FM Ratio | 6.38 (0.72) | 6.38 (0.72) | 6.43 (0.65) |
| Arm Muscle Area (mm$^2$) | 1655.0 (403.79)[a] | 1775.4 (482.47)[ac] | 1801.4 (341.30)[bc] |
| Arm Fat Area (mm$^2$) | 365.34 (148.74)[a] | 370.50 (136.21)[ac] | 320.34 (114.41)[bc] |

N.B. Each row with different superscript is significantly different, BMI = Body Mass Index

FFMI = Fat Free Mass Index, FMI = Fat Mass Index, MUAC = Mid Upper Arm Circumference

In contrast, height was not significantly different among groups three, four, and five. The average height of the children aged 2–5 years was around 96 cm, whereas this measurement was 152 cm for children aged 12–15 years. MUAC of children was similar when comparing groups three and four and between groups four and five. At the same time, the other intergroup comparison is shown to have a significant difference (p≤0.05). Similarly, BMI and FMI were found to be similar when comparison was made among groups one, two and three, while another comparison was found significant (p≤0.05).

Our study follows the traditional two-compartment model to measure body composition. Fat Mass (FM) and Fat-Free Mass were measured from skinfold thickness and BMI. The fat mass was found similar only when the comparison is made between groups four and five; a significant difference was observed when the comparison is made among other groups. However, the percent of body fat content was similar to all the five age groups (p>0.05). Like percent body fat, the fat-free mass ratio to fat mass was also similar among groups at around 6% (p>0.05). The highest average arm muscle area (2275.5 mm$^2$) was reported for children between the ages group 12 to 15 years. The arm muscle area was significantly different among different age groups, except only between groups three and four and groups four and five. In contrast, the arm fat area was only significantly different when comparing groups two and five. The average sum of the four-skinfold thickness in the age group of 8–10 years was the highest (14.39mm); in contrast, the lowest 12.76 mm was reported for children in the age category 10–12 years. According to age category, triceps and biceps skinfold thickness was reported to be similar among five age groups (p>0.05), while other skinfold thickness was significantly

**Table 3. Distributions of infant characteristics stratified by age category (n = 330).**

| Characteristic | 2–5 Years | 5–8 Years | 8–10 Years | 10–12 Years | 12–15 Years |
|---|---|---|---|---|---|
| | (n = 72) | (n = 139) | (n = 96) | (n = 10) | (n = 13) |
| | mean (SD) | Mean (SD) | mean (SD) | mean (SD) | Mean (SD) |
| Age (Months) | 46.44 (10.68) | 82.19 (9.93) | 109.89 (8.02) | 138.00 (6.34) | 162.46 (9.32) |
| Weight (kg) | 13.53 (3.72)$^a$ | 19.68 (5.15)$^b$ | 24.44 (5.51)$^c$ | 34.35 (8.29)$^d$ | 47.39 (10.75)$^d$ |
| Family Income (BDT) | 8345.17 (4173.56)$^a$ | 9645.32 (9795)$^a$ | 8682.29 (5098.05)$^a$ | 135000 (12249)$^a$ | 1230769 (14103.1)$^a$ |
| Height (cm) | 96.79 (11.40)$^a$ | 115.25 (10.39)$^b$ | 129.17 (9.00)$^c$ | 140.0 (7.20)$^c$ | 152.52 (12.66)$^c$ |
| MUAC (mm) | 147.26 (11.54)$^a$ | 157.71 (17.69)$^b$ | 170.04 (14.09)$^c$ | 176.00 (1174)$^{cd}$ | 183.46 (16.38)$^d$ |
| *Skin-fold Thickness (mm)* | | | | | |
| Triceps | 4.75 (1.45)$^a$ | 4.28 (1.44)$^a$ | 4.51 (1.51)$^a$ | 3.9 (0.63)$^a$ | 4.74 (1.03)$^a$ |
| Biceps | 3.04 (0.81)$^a$ | 3.00 (1.10)$^a$ | 2.98 (1.25)$^a$ | 3.33 (1.16)$^a$ | 3.69 (0.93)$^a$ |
| Subscapular | 3.78 (1.27)$^{ae}$ | 3.26 (1.32)$^{bd}$ | 4.09 (1.82)$^{ce}$ | 3.14 (0.74)$^{de}$ | 3.04 (1.05)$^{de}$ |
| Suprailiac | 2.66 (0.89)$^{ae}$ | 2.28 (0.84)$^{bd}$ | 2.80 (1.10)$^{ce}$ | 2.40 (0.47)$^{de}$ | 2.46 (0.62)$^{de}$ |
| *Sum of Skinfold* | 14.22 (3.49)$^{ae}$ | 12.82 (4.01)$^{bd}$ | 14.39 (4.21)$^{ce}$ | 12.76 (2.28)$^{de}$ | 13.93 (2.52)$^{de}$ |
| *Thickness (mm)* | | | | | |
| BMI (kg/m$^2$) | 14.51 (3.35)$^a$ | 14.80 (3.47)$^a$ | 14.52 (2.40)$^a$ | 17.34 (2.89)$^b$ | 20.20 (3.34)$^c$ |
| FFMI (Kg/m$^2$) | 12.52 (2.95)$^a$ | 12.81 (2.99)$^{ab}$ | 12.52 (2.06)$^a$ | 15.00 (2.60)$^b$ | 17.33 (2.78)$^c$ |
| FMI (kg/m$^2$) | 1.99 (0.44)$^{ab}$ | 1.99 (0.55)$^{ab}$ | 2.00 (0.42)$^{ab}$ | 2.34 (0.34)$^b$ | 2.86 (0.60)$^c$ |
| Fat Mass (kg) | 1.86 (0.49)$^a$ | 2.66 (0.87)$^b$ | 3.38 (0.93)$^c$ | 4.63 (1.04)$^{de}$ | 6.75 (1.92)$^e$ |
| Percent Body Fat | 13.78 (1.16)$^a$ | 13.43 (1.46)$^a$ | 13.75 (1.48)$^a$ | 13.55 (1.08)$^a$ | 14.10 (1.03)$^a$ |
| Fat Free Mass (kg) | 11.67 (3.26)$^a$ | 17.02 (4.37)$^b$ | 21.06 (4.68)$^c$ | 29.72 (7.31)$^c$ | 40.59 (8.89)$^c$ |
| FFM/FM Ratio | 6.31 (0.62)$^a$ | 6.52 (0.69)$^a$ | 6.35 (0.73)$^a$ | 6.42 (0.58)$^a$ | 6.13 (0.50)$^a$ |
| Arm Muscle Area (mm$^2$) | 1403.28 (235.20)$^a$ | 1676.28 (343.62)$^b$ | 1946.15 (308.52)$^c$ | 2144.27 (320.46)$^{cd}$ | 2275.50 (361.12)$^d$ |
| Arm Fat Area (mm$^2$) | 332.81 (107.16)$^{ab}$ | 327.71 (144.76)$^a$ | 370.39 (138.94)$^{ab}$ | 330.59 (52.79)$^{ab}$ | 422.64 (123.34)$^b$ |

N.B. Each row with different superscript is significantly different, BMI = Body Mass Index, FFMI = Fat Free Mass Index, FMI = Fat Mass Index, MUAC = Mid Upper Arm Circumference.

different among different age groups. The lowest skinfold thickness measurement was recorded for suprailiac skinfold (2.28–2.80 mm), whereas the highest was recorded for triceps skinfold (3.9–4.75 mm).

Figs 2–4 show changes in body composition during the growing period of children. BMI, FFMI, and FFM (kg) remain stable over the period of 2 to 10 years with slight fluctuations and

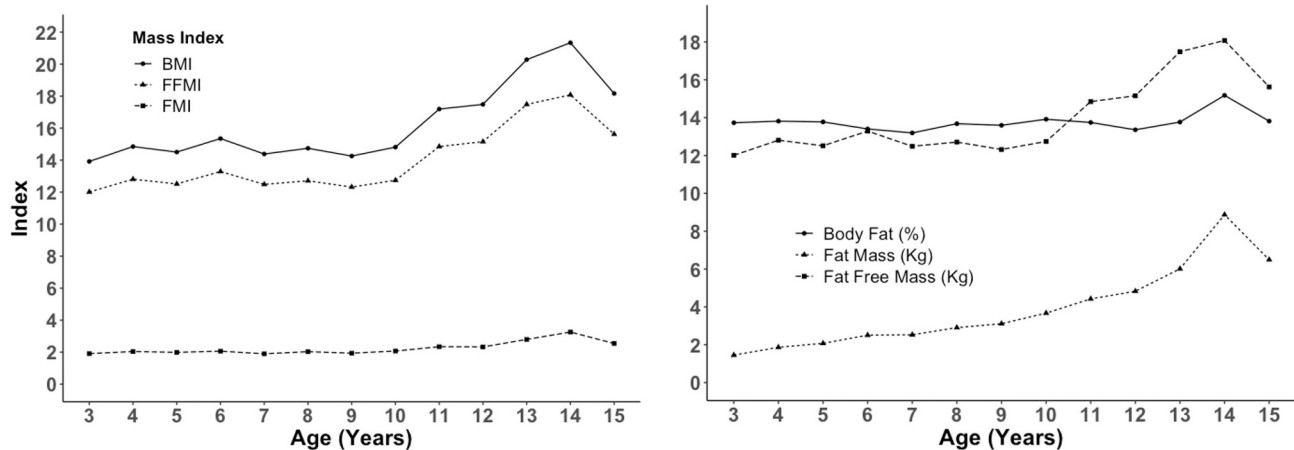

**Fig 2. Trajectories of BMI, FFMI and FMI plotted against age in year (n = 330).**

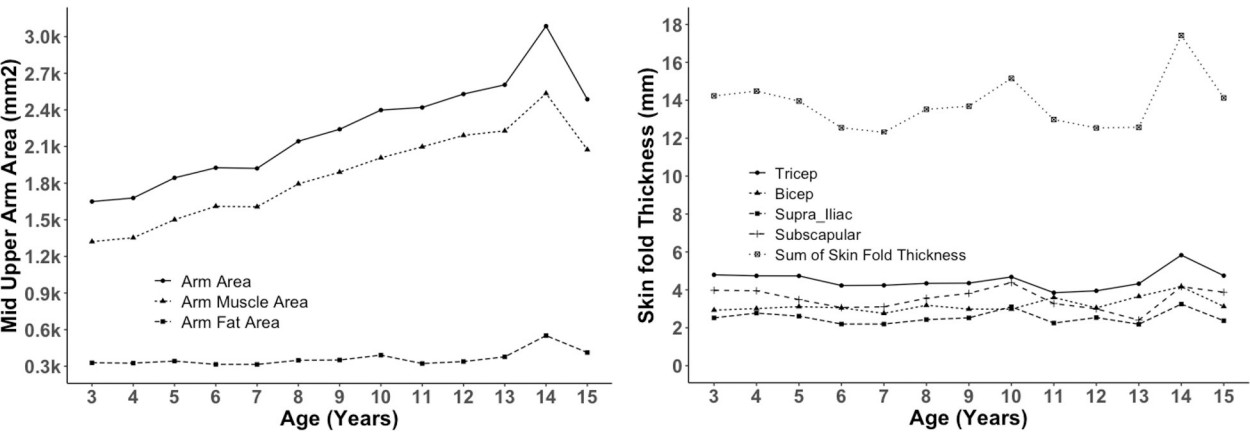

**Fig 3. Trajectories of % body fat, fat mass, fat free mass and sum of skinfold thickness plotted against age in year (n = 330).**

then rise slowly, reach a peak at age 14, and then decrease again at 15 years, while FMI and percent body fat did not change over the period. These are primarily attributed to the increase in fat mass and fat-free mass (Fig 2). Similar to FMI, percent body fat also remains steady over the period of 2 to 15 years. FFM increases gradually from 2 Kg at three years to around 6 kg at 13 years, reaches the peak of 8 kg at 14 years, and then returns to 6 kg at 15 years (Fig 2). Arm Area, Arm Muscle Area, and Arm Fat Area also follow the similar trend of Mass Index (Fig 3). The sum of four skinfold thickness increases slightly with slight fluctuation before reaching a peak of around 18 at 14 years, then decreases (Fig 3). The peak of the sum of skinfold thickness at 14 years can also be correlated with the peak of fat mass (Fig 2) and arm area (Fig 3). The percent of body fat for both boys and girls were similar and unchanged throughout adulthood, whereas FFM (kg) increased gradually for both sexes (Fig 4).

## Children's nutritional status

Table 4 shows the percentage of children classified as malnourished according to height-for-age, weight-for-height, weight-for-age indices, and BMI-for-Age, by age, sex, and study area. The data show that around 31% of the total study children (2–10 years) were considered underweight (low weight-for-age), approximately 15% were severely underweight. The mean Weight-for-age z score was reported -1.295, and the highest prevalence of underweight was

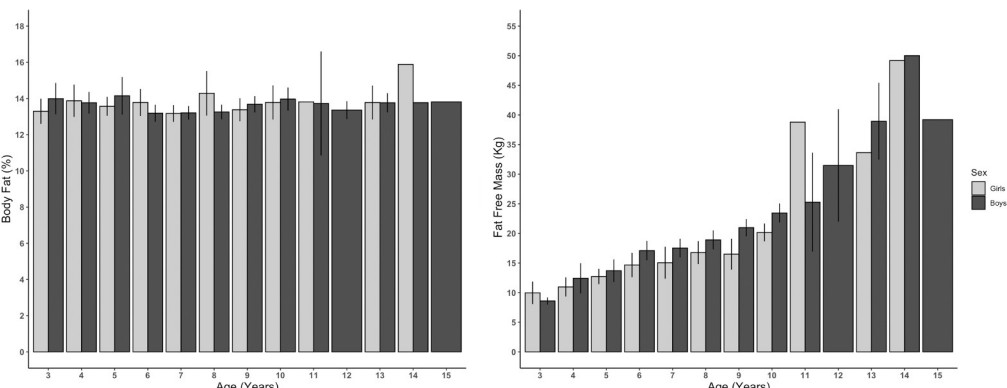

**Fig 4. Percent body fat and fat free mass of boys (n = 208) and girls (n = 122) by age category (man±SE) (n = 330).**

**Table 4. Percentage of children aged 2–15 years classified as malnourished according to anthropometric indices of nutritional status: Height-for-age, weight-for-height, weight-for-age and BMI-for-age.**

| Age groups (Months) | | Weight-for-age %* (Underweight) | | | Length/height-for-age % (Stunted) | | | Weight-for-length/height %* (Wasted and Overweight) | | | | | BMI-for-age % (Wasted and Overweight) | | | | |
|---|---|---|---|---|---|---|---|---|---|---|---|---|---|---|---|---|---|
| | N (%) | % (n) < -3 SD | % (n) < -2SD | Mean (95% CI) | % (n) < -3SD | % (n) < -2SD | Mean (95% CI) | % (n) < -3SD | % (n) < -2SD | % (n) > +2SD | % (n) > +3SD | Mean (95% CI) | % (n) < -3SD | % (n) < -2SD | % (n) > +2SD | % (n) > +3SD | Mean (95% CI) |
| *2 to 15 years* *(24–180 Months)* | 330* (100) | 15.3 (47) | 16.6 (51) | -1.295 (-1.113 to -1.478) | 9.4 (31) | 15.5 (51) | -0.937 (-0.738 to -1.136) | 20 (36) | 11.1 (20) | 6.1 (11) | 8.9 (16) | -0.842 (-0.463 to -1.220) | 15.2 (50) | 13.6 (45) | 5.5 (18) | 3.9 (13) | -1.009 (-0.778 to -1.241) |
| *2 to 5 Years* *(24–60)* | 72 (21.82) | 19.44 (14) | 18.06 (13) | -1.494 (-1.103 to -1.885) | 18.06 (13) | 16.67 (12) | -1.194 (-0.623 to -1.765) | 17.39 (12) | 14.49 (10) | 1.45 (1) | 8.7 (6) | -1.087 (-0.450 to -1.674) | 18.06 (13) | 15.28 (11) | 4.17 (3) | 8.33 (6) | -1.049 (-0.455 to -1.645) |
| *5 to 10 Years* *(61–120)* | 235 (235) | 14.04 (33) | 16.17 (38) | -1.234 (-1.028 to -1.441) | 6.81 (16) | 15.32 (36) | -0.860 (-0.646 to -1.074) | 21.62 (24) | 9.01 (10) | 9.01 (10) | 9.01 (10)) | -0.689 (-0.190 to -1.188) | 15.74 (37) | 14.04 (33) | 5.96 (14) | 2.98 (7) | -1.101 (-0.838 to -1.364) |
| 5 to 8 Years (61–96 Months) | 139 (42.12) | 17.27 (24) | 13.67 (19) | -1.204 (-0.910 to -1.497) | 8.63 (12) | 18.71 (26) | -0.973 (-0.655 to -1.290) | 20.83 (20) | 9.38 (9) | 9.38 (9) | 8.33 (8) | -0.648 (-0.109 to -1.186) | 16.55 (23) | 9.35 (13) | 7.19 (10) | 5.04 (7) | -0.922 (-0.535 to -1.308) |
| 8 to 10 Years (97–120 Months) | 96 (29.09) | 9.38 (9) | 19.79 19 | -1.279 (-0.999 to -1.559) | 4.17 (4) | 10.42 (10) | -0.697 (-.0.441 to -0.953) | 26.67 (4) | 6.67 (1) | 6.67 (1) | 13.33 (2) | -0.954 (0.532 to -2.440) | 14.58 (14) | 20.83 (20) | 4.17 (4) | 00 (00) | -1.361 (-1.041 to -1.681) |
| *10 to 15 Years* *(121–180 Months)* | 23* (6.97) | -- -- | -- -- | -- -- | 8.70 (2) | 13.04 (3) | -0.916 (-0.368 to -1.464) | -- -- | -- -- | -- -- | -- -- | -- -- | 00 00 | 4.35 (1) | 4.35 (1) | 00 00 | 0.051 (0.625 to -0.522) |
| 10 to 12 Years (121–144 Months) | 10 (3.03) | -- -- | -- -- | -- -- | 00 (00) | 10.00 (1) | -0.913 (-0.295 to -1.531) | -- -- | -- -- | -- -- | -- -- | -- -- | 00 (00) | 00 (00) | 00 (00) | 00 (00) | -0.268 (0.717 to -1.253) |
| 12 to 15 Years (141–180 Months) | 13 (3.94) | -- -- | -- -- | -- -- | 15.38 (2) | 15.35 (2) | -0.918 (0.014 to -1.851) | -- -- | -- -- | -- -- | -- -- | -- -- | 00 (00) | 7.69 (1) | 7.69 (1) | 00 (00) | 0.297 (1.074 to -0.480) |

N.B.

* Weight-for-age and weight-for height reference data are not available beyond age 10

reported among children aged 2 to 5 years, and around 2/5th of children was reported to be underweight. In contrast, almost half of them were severely underweight. According to the height-for-age z score, around a quarter of children were reported to be stunted, and the mean height-for-age z score was -0.937. Similar to underweight, the highest prevalence of severely stunted and moderately stunted children was reported to be within the age range of 2 to 5 years (Table 4). Around 30% of children were reported to be wasted according to the BMI-for-age z score, and the mean z score of all children (n = 330) was -1.009, and the highest prevalence was also reported to be among children aged 2 to 5 years. Only 13 children were found to be obese and 18 as overweight (Table 4). All sort of malnutrition was higher among children

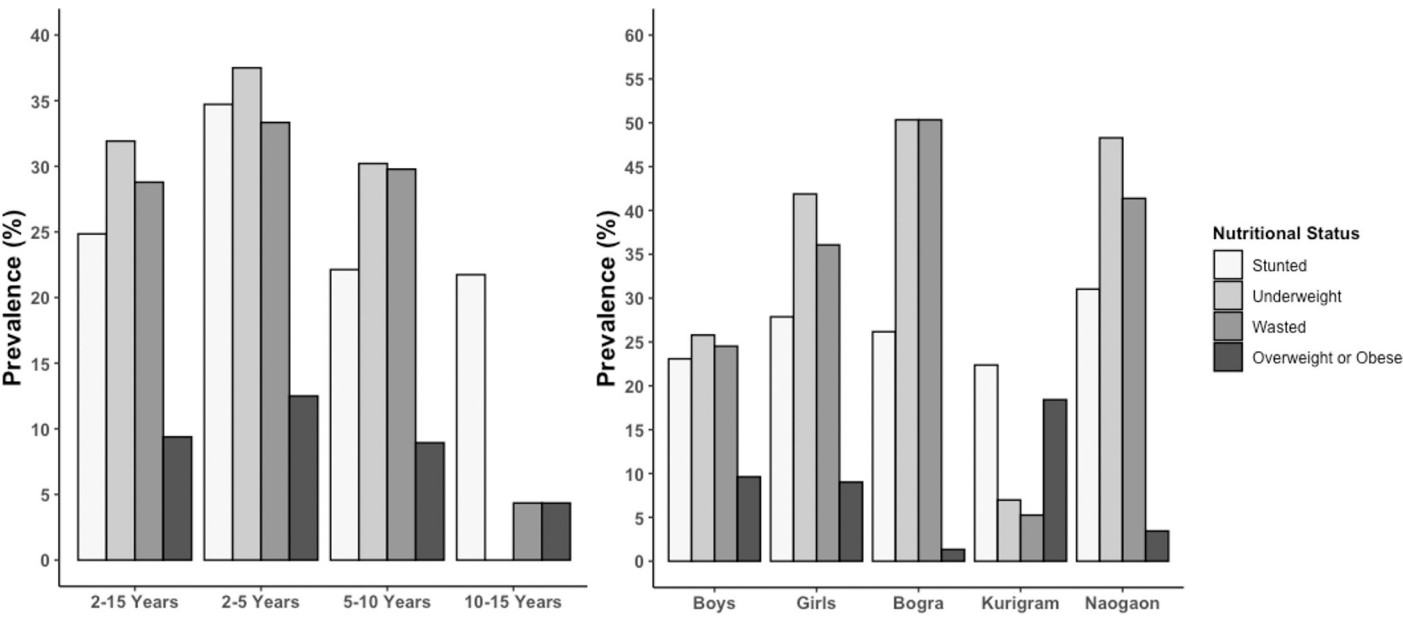

**Fig 5. Trends of nutritional status of children aged 2–15 years (n = 330).**

aged 2 to 5 years, and the prevalence was higher among girls. The prevalence of wasted and underweight was higher among children from Bogra, while the prevalence of stunting was slightly higher among children from Naogaon. Prevalence of overweight and obesity was reported to be higher among children from Kurigram (Fig 5).

The median BMI of studied children was plotted against age and compared with WHO 2007 reference standard. The median BMI of studied male children was reported within the range of 15th to 25th percentile of WHO reference standard over the period of 2 to 13 years and at 13 reaches 75th percentile and reached again below 50th percentile. The median BMI of girls remained below the 15th percentile of reference standard up to 10 years, then soar over the 85th percentile and stayed within 75th to 85th percentile (Fig 6). When height was compared

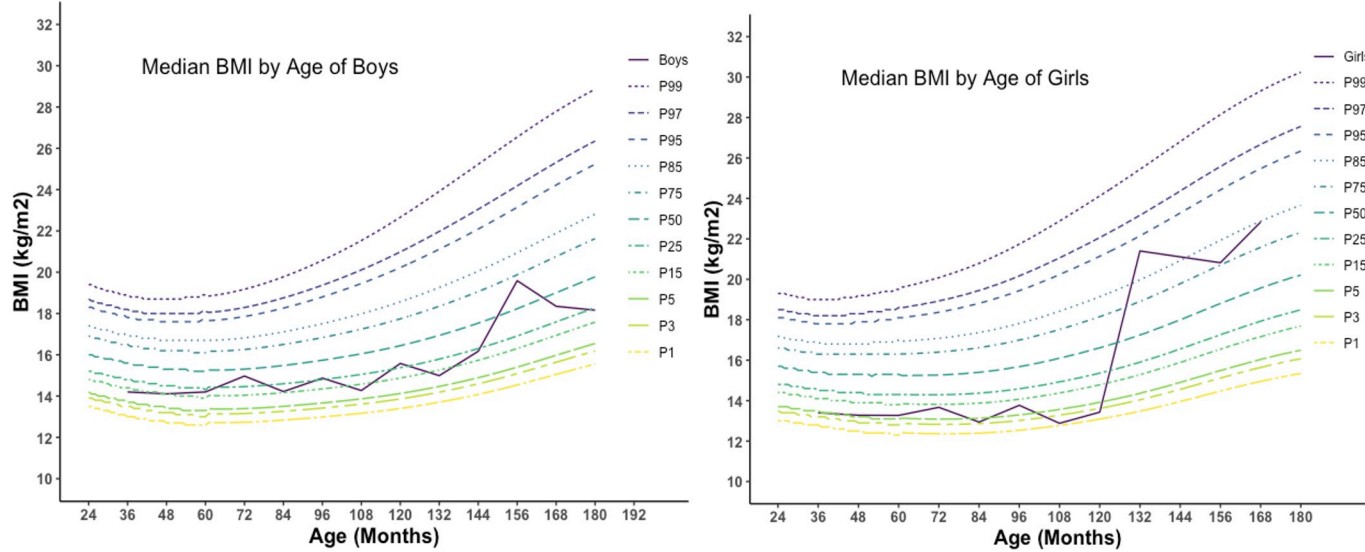

**Fig 6. BMI for age percentile for boys and girls (n = 330) [Graphs was generated using R].**

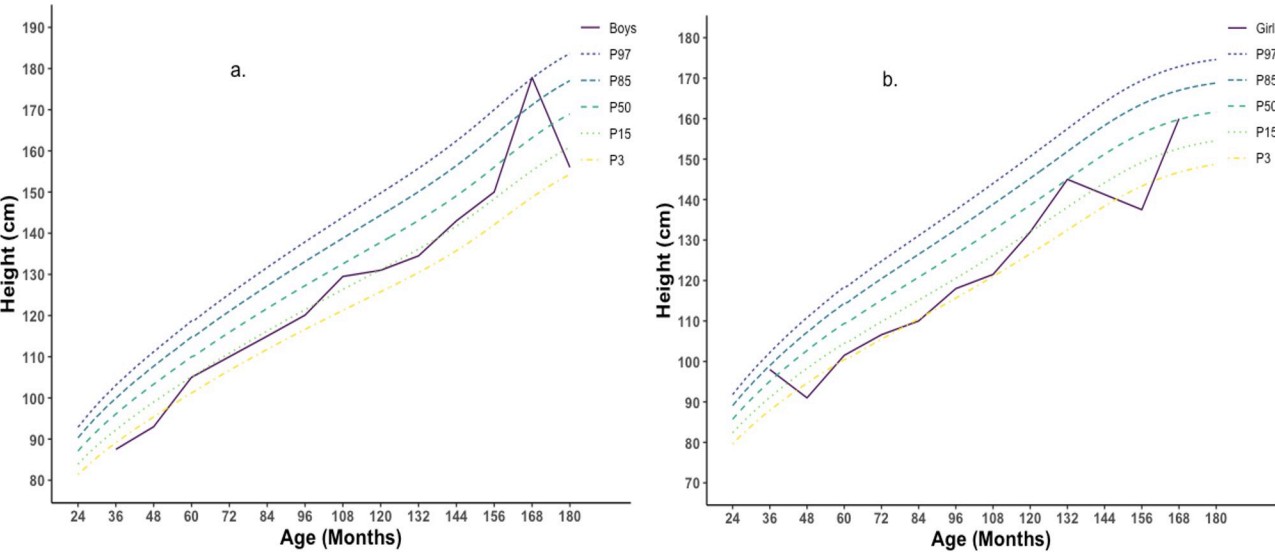

**Fig 7. Height for age percentile for Boys (a.) and Girls (b.) (n = 330) [Graphs was generated using R].**

with WHO 2007 reference standard, the median height of boys and girls was reported to be within the 15[th] percentile of the reference standard with slight fluctuations (Fig 7).

Binary logistic regression analysis showed that girls were two times more likely to be wasted compared to boys. Children who have handwashing practice are less likely to be underweight (OR = 3.53), wasted (OR = 2.31), and stunted (OR = 2.72). Children whose mothers were service holders were less likely to be wasted than to children whose mothers were housewives. Children from lower family income quartile were more likely to be stunted (OR = 3.16), and less likely to be wasted (OR = 0.423) and obese (OR = 0.177) compared to the higher income quartile family (Table 5). The average weight-for-age z score and height-for-age z score were plotted according to the family income quartile (Fig 8). Children from lower-income quartiles were shown to have lower average SD scores. The binary logistic regression also supports this finding.

## Discussion

Every method has an assumption, and the assumption used in each method does not grasp proper in all cases. Usually, a combination of measurements is used to derive the best model from minimizing such assumptions [33]. Our study involves an anthropometric analysis of nutritional status based on two-compartment body composition models. A heterogeneous group of children from three different northern districts with both gender, different family income levels, and age groups (2–15 years) was involved. The overall nutritional status of children from the current study was below the reference standard, and family income, mother's job, handwashing practice, and gender were the strong predictors.

During adulthood, because of progressive ontogenesis, a firm increase in fat mass and fat free mass due to body growth, is observed [34, 35]. FFMI and FMI as a new concept have been described previously for adults and elderly individuals and used as an indicator of nutritional status [25, 36]. Like BMI, FFMI from the current study also follows a similar increasing trend over the period, except for FMI. FM and FFM are usually expressed either as percentages (%) or in the absolute unit (kg), which is unsatisfactory [25, 37]. For example, a tall child, who suffers from undernutrition, can exhibits values for FM and FFM similar to those of a shorter

**Table 5. Factors associated with malnutrition in using multivariate binary and polynomial logistic regression model.**

| | Weight-for-age | Length/height-for-age | BMI-for-age | |
| --- | --- | --- | --- | --- |
| | Underweight | Stunted | Wasted | Overweight |
| | OR | OR | OR | OR |
| | (95% CI) | (95% CI) | (95% CI) | (95% CI) |
| **Sex** | | | | |
| Male (Ref) | 1 | 1 | 1 | 1 |
| Female | 1.951 (1.150–3.311)* | 1.281 (0.734–2.220) | 1.55 (0.894–2.68) | 1.253 (0.54–2.909) |
| **Age** | | | | |
| 2–5 Years (Ref) | 1 | 1 | 1 | 1 |
| 5–10 Years | 0.991 (0.536–1.833) | 0.579 (0.312–1.074) | 1.11 (0.578–2.139) | 0.614 (0.2462–1.533) |
| 10–15 Years | -- | 0.699 (0.212–2.297) | 0.118 (0.0127–1.107) | 0.163 (0.0182–1.461) |
| **Drinking Water** | | | | |
| Tube-Well (Ref) | 1 | 1 | 1 | 1 |
| Tank Water | 0.725(0.165–3.176) | 1.698 (0.370–7.778) | 0.311 (0.069–1.394) | 1.246e-09 (1.246e-09–1.246e-09)*** |
| Tank + Tube-well | 4.061e-07(0.000 –Inf) | 6.84 (NA–Inf) | 3.49e-07 (3.49e-07–3.49e-07)*** | 2.971e-07(2.971e-07–2.917e-07)*** |
| **Hand Wash Habit** | | | | |
| Yes (Ref) | 1 | 1 | 1 | 1 |
| No | 3.531(1.657–7.525)** | 2.311 (1.081–4.940)* | 2.725 (1.256–5.910)* | 0.6113(0.124–2.99) |
| **Mother Education** | | | | |
| Literate (Ref) | 1 | 1 | 1 | 1 |
| Illiterate | 0.716(0.405–1.267) | 0.736 (0.413–1.311) | 1.053 (0.589–1.879) | 1.425(0.613–3.308) |
| **Mother Job** | | | | |
| Housewife (Ref) | 1 | 1 | 1 | 1 |
| Labour | 3.12(0.274–35.529) | 0.445 (0.043–4.542) | 1.827 (0.254–13.10) | 1.318e-06(1.318e-06–1.318e-06)*** |
| Service Holder | 4.56e-07 (0.00 –Inf) | 8.233–07 (NA–Inf) | 5.149e-07 (5.14e-07–5.14–07)*** | 8.211e-07(8.211e-07–8.211e-07)*** |
| **Family Income** | | | | |
| Quartile 1 | 1.308 (0.539–3.172) | 3.162 (1.201–8.323)* | 0.423 (0.173–0.984* | 0.446 (0.158–1.261) |
| Quartile 2 | 2.148 (0.882–5.229) | 2.008 (0.728–5.533) | 1.195 (0.511–2.793) | 0.177 (0.0448–0.702)* |
| Quartile 3 (Ref) | 1 | 1 | 1 | 1 |
| Quartile 4 | 1.919 (0.505–7.285) | 0.896 (0.162–4.958) | 0.902(0.245–3.308) | 0.341 (0.0355–3.286) |

N.B.

*p≤0.05

**p≤0.001

*** p≤0.0001; Family Income Quartile 1 = <7000 BDT, 2 = 7000–11999 BDT, 3 = 12000–20000 BDT, 4 = >20000.

well-nourished child. Height normalized indexes like Fat Mass Index (FMI) and Fat Free Mass Index (FFMI) could avert these difficulties. The observed findings of FFMI and FMI of boys and girls from the current study were slightly lower than the findings of Nakao, T. et al., (2003); however, the average increase of FFMI and FMI was much lower [37]. The average MUAC of our studied children, aged 2–5 years, was within the standard cutoff point (>13mmm) [26]. The average MUAC of the other four age groups and the average MUAC of boys and girls and in the three different dwelling areas also fulfil this cutoff point [26, 38]. However, there are limited data that correlate directly MUAC with other body fat and malnutrition measures, and this index has significant variability in measurement and needs standardization. Our study used both skinfold thickness and BMI to measure fat mass, fat-free mass, and % body fat. However, Astrid CJ Nooyens et al. (2007) suggests skinfold thickness over BMI to measure body fatness [39] and encourage the subscapular site to choose as the

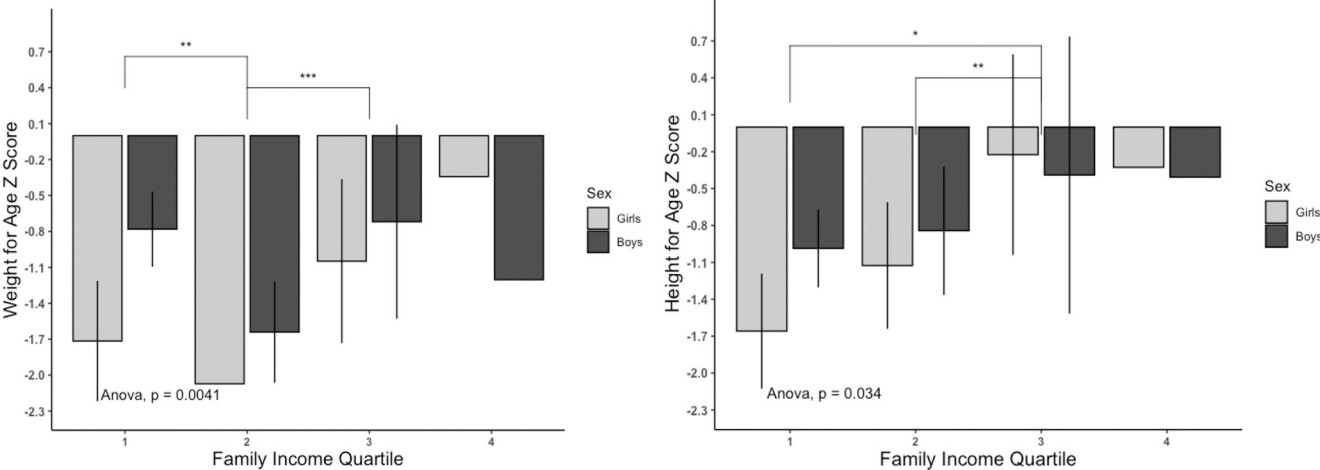

**Fig 8. Weight-for-age Z score and height-for-age Z score of children of age 2–15 years plotted against family income quartile (n = 330) [N.B.: Family income quartile 1 = <7000 BDT, 2 = 7000–11999 BDT, 3 = 12000–20000 BDT, 4 = >20000].**

choice of the best site. Our findings show an inconsistent result for the sum of 4 skinfold sites, as it fluctuates abnormally over the period of 2–15 years. Four skinfold thickness measurements were included in the current study; however, to the best of our knowledge there is no study to include four sites for skinfold thicknesses measurement sites. At ten years, Ahmad M et al. observed the mean triceps skinfold thickness of 7.20 mm, which was higher than our findings (4.51 mm) [40].

In our study, an increase in fat mass (kg) and fat-free mass (kg) was observed during adulthood; however, percent body fat remained within the range of around 14%. This trend may be described because the total fat mass and fat-free mass increase as the child grows, while percentages remain steady. Analysis of body composition of 7–10 years Bangladeshi children by Khan et al. (2012) reported slightly higher body fat percentages (15.89±5.87) than the current study (13.63±1.39). However, the same study reported much-lowered rates of body fat (7.81 ±3.31) using the Tanita system [41]. Though DEXA is considered the gold standard for body composition analysis [15], the isotope dilution technique is also accurate and precise [42]. As our finding is in close agreement with the isotope dilution technique, this study's findings could be considered reliable. Our study reveals that fat-free mass increases gradually and peaks at around 18 kg at 14, starting at almost 14 kg at 2–3 years. This finding is lower than the findings of Cynthia L. Ogden et al. (2011), and the study of Cynthia L. Ogden et al. states that male children gain fat-free mas and female children gain more at the onset of puberty fat than males [43]. A higher fat mass among boys than girls and higher % body fat in girls than boys were observed in the current study. However, Soledad Aguado-Henche et al. (2011) observed no gender-specific difference [44]. However, Jaydip Sen & Nitish Mondal (2013) reported a sex-specific significant difference in FM and FFM among children 5–12 years in West Bengal, India [29].

Additionally, according to the age category, our findings of total fat mass and % body fat were much lower than Soledad Aguado-Henche et al. (2011). However, Henche et al. (2006) also observed an increase in lean mass of females until age 15; after then it stabilizes till the age 80 years [45]. Therefore, the significantly higher amount of average fat mass, FMI, and Arm Muscle Area among children from the district of Kurigram than Bogra and Naogaon may be due to a better dietary intake.

From the average SD score for weight-for-age, height-for-age, weight-for-length, and BMI-for-age, the studied children were undernourished, and a small proportion of children was considered Obese (3.9%) and overweight (5.5%). The percentage of underweight and low BMI-for-age (wasted) children was higher at all age categories (25 to 35%) and was found very high compared to the BDHS report, where the prevalence of stunting, underweight, and wasting in Bangladesh was 31, 22, and 8% respectively [10]. Stunting was reportedly lower in the current study than the BDHS 2017–18 report, while the other two forms of undernutrition were higher. Nisbett, Nicholas, et al. (2017) reported a declining trend of population undernourishment from the 1990s and plateaued by the mid-2000s due to extreme poor in different regions in Bangladesh [46]. The high prevalence of wasting and underweight from the current study in the northern area of Bangladesh also supports the findings of Nisbet, Nicholas, et al. (2017). The prevalence of wasting was higher than the WHO cut-off point ($\geq$15%) [47, 48], and the highest prevalence (~50%) was observed in the district of Bogra.

In contrast, in Naogaon and Kurigram, this percentage was around 42% and 5%, respectively, and in Naogaon, the prevalence was more than the cut-off value. All forms of undernutrition were lower among children from Kurigram, even lower than the recent BDHS report, and the Overweight/Obesity level was the highest in this district. The findings of the lower level of undernourishment in Kurigram are also reflected in the findings of body composition data. The percentage of children with height-for-age SD values below -2SD was 20 to 30% which indicates a medium degree of malnutrition. Weight-for height indices measure the current nutritional status of children, whereas Height-for-age shows the cumulative linear growth and is influenced by long-term nutritional deficiencies. The highest prevalence of undernourishment and overweight was reported among children of age group 2–5 years and decreases as the age increases. However, few studies reported a similar finding for same age group children in Bangladesh to compare [49, 50]. This higher prevalence among children 2–5 years could be explained as improper or insufficient weaning food or nutrition during this age. According to the India National Family Health Survey (NFHS-4), 2015–16 prevalence of underweight, stunted and wasted were similar to findings from the current study [51]. The similar prevalence could be explained as the similar sociodemographic and economic setting of the two countries. The prevalence of underweight and wasting of under-five children from the current study was much higher than in Ethiopia, while the prevalence of stunting was slightly lower [52]. The degree of undernutrition could also be noticed from BMI-for-age (Fig 6) and Height-for-age (Fig 7) percentile figure, and the median BMI-for-age and height-for-age were below the WHO 2007 reference standard. The overall prevalence of stunting, wasting, and underweight was higher among girls than boys (Fig 5), which is agrees with the World Bank report for Bangladesh [53]. This higher prevalence among girls may be due to poor dietary diversity and inequality in intra-household food distribution that girls suffer most, which might be true for children of the current study.

Binary and polynomial logistic regression was performed to explore few potential factors of malnutrition: gender, hand washing practice as a factor for underweight; handwashing habit and family income for stunting and source of drinking water, mother job and family income for wasting were predicted to be responsible. Like a higher prevalence of underweight among girls, logistic regression also shows almost two times higher odds of being underweight than boys. The higher odd among girls may be due to disparity in household food distribution or gender discrimination. Children who don't practice hand washing are more prone to suffering from underweight and stunting than those who practices hand washing. The World Health Organization (WHO) appraises that around half of the cases of child undernutrition are related to recurrent diarrhea and related disorder, which are directly linked to hygiene,

sanitation, and handwashing habits [54]. Several studies also directly linked hand hygiene and washing practice with child undernutrition [55–58].

Family income was found to be linked to wasting and stunting. From logistic regression, children from a family with the lowest (1st) income quartile were three times more prone to suffering from stunting (OR:3.162, CI:1.201–8.323); however, the odd (OR:0.423, CI:0.173–0.984) of wasting was lower among this group of children when compared to children from 3rd family income quartile. Furthermore, the average weight-for-age and height-for-age z scores were lowest among the lowest family income quartile group and as the family income rises, the z score increases (Fig 8). This relation indicates an apparent link between family income and being underweight and wasting. The stunting results from long-term nutritional deficiency and family income may affect the height-foe-age z score directly or indirectly through food accessibility and food diversity. In contrast, the odds of being wasting is reported to be lowered among low family income group children. This lower odd of wasting may be explained as a family with low income rely on energy-dense, low diversified food for their children, which may have a more negligible effect on long term (stunting) instead of short-term nutritional status. Perhaps families with low-income may not access or afford proper nutrition for their children, which would be crucial for the long-term growth and development of their children. Several studies on children in Bangladesh also supported that low family income negatively impacts underweight and stunting [49, 50, 59, 60]. Studies from Iran and Maldives also support similar findings that the odds of stunting, wasting, and underweight children in low family income groups are higher than those of the higher-income quintile family group [61, 62].

## Strength and limitations

To the best of our knowledge, the current study is the first study of nutritional status in a combination of body composition analysis. Most of the studies done in Bangladesh focused on anthropometry and nutritional status. Therefore, the chief strength of this study is that body composition data have been combined to validate nutritional status. This validation has been reflected in the data of body composition and nutritional status. The second strength of this study is that both SD score and percentile ranks have been used to compare with the WHO 2007 reference standard. Despite these strengths, there are few limitations of the two-compartment body composition model that has been used in the current study. First, though the two-compartment model is cost-effective and straightforward, it is subject to error since an assumption is used to measure body fat and fat-free mass. The constant factor used in the equation may not be accurate and precise, which possesses some limitations. Second, the small sample size and the disproportionate number of samples among different ages, gender, dwelling, and family income categories may entail regression and other statistical analysis problems. Third, family income is an essential factor of undernutrition; however, family income has been measured in absolute terms, and most of the time the arbitrary oral response has been recorded. Indirect and other forms of family income may have been excluded by the responder, which may impact the outcome. Fourth, food security and food diversity, and dietary record data have not been collected, which may have an important impact on undernutrition, particularly in the long run.

## Conclusions

In conclusion, our study reveals that the nutritional status of children and adolescents, based on anthropometric and body composition analysis, was below the reference standard. Despite efforts that the Government and NGOs have paid, improvements have been made to reduce malnutrition in Bangladesh. The rate of underweight, wasting, and stunting is high among

children of the three northern districts of Bangladesh. The average Z score for weight-for-age, height-for-age, and BMI-for-age was negative and below the reference standard. Children within the age groups 2–5 years and girls were more vulnerable. The percentile rank of the studied children was also below the reference standard. The potential factor may be responsible for the higher rate of undernutrition found includes gender, hand washing practice, source of drinking water and family income. Therefore, to reduce the rate of malnutrition among children within an acceptable range, an integrated programme involving the government, non-governmental organizations, and the community is undeniably necessary. A gender-specific nutrition intervention programme should be implemented targeting girls. Hand washing, potable drinking water, and proper sanitation and hygiene facilities targeting the children of the northern area of Bangladesh should be implemented. Nutrition sensitive and nutrition-specific programmes to raise direct family income, like cash transfer, creation of income-generating activities, could be a practical option. Moreover, working diligently with local communities through cooking demonstrations and food fairs to build understanding and ensuring seasonal food availability through the promotion of family gardening can ensure the availability of diverse food throughout the years.

Further studies with more sample size and a proportionate number of samples from different categorical variables might be helpful in better understanding the children's nutritional status and their predicting factors. In addition, it is suggested to include a four-compartment body composition model with gold standard technique in the future, which would be likely to explore child nutritional status with more reliable results.

## Supporting information

**S1 File. Survey questionnaire.**
(PDF)

## Acknowledgments

The authors thank the study participants, family members and caregivers of the participants, chairman and member of the Union Parishad, and teachers at local schools. The authors also thank Tawhid Hossain, Doctoral Researcher, Leibniz-Center for Agricultural Landscape Research (ZALF), Germany, for his help preparing the study area map. Finally, the authors are also thankful to Md Rezaul Haque, PhD, English Department at St. John's University, New York, for his editorial help.

## Author Contributions

**Conceptualization:** Md. Kamruzzaman.

**Data curation:** Md. Kamruzzaman, Shah Arafat Rahman, Sharmin Akter, Humaria Shushmita, Md. Yunus Ali, Md Adnan Billah, Md. Sadat Kamal.

**Formal analysis:** Md. Kamruzzaman.

**Methodology:** Md. Kamruzzaman.

**Software:** Md. Kamruzzaman.

**Validation:** Md. Kamruzzaman, Dipak Kumar Paul.

**Visualization:** Md. Kamruzzaman.

**Writing – original draft:** Md. Kamruzzaman.

**Writing – review & editing:** Md. Kamruzzaman, Shah Arafat Rahman, Humaria Shushmita, Md. Yunus Ali, Md Adnan Billah, Md. Sadat Kamal, M. Toufiq Elahi, Dipak Kumar Paul.

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
