## [Decision Letter · Decision Letter 0]

11 Jun 2021

PONE-D-21-12299

The anthropometric assessment of body composition and nutritional status in children aged (2-15 years): A cross-sectional study from three districts in Bangladesh

PLOS ONE

Dear Dr. Kamruzzaman,

Thank you for submitting your manuscript to PLOS ONE. After careful consideration, we feel that it has merit but does not fully meet PLOS ONE’s publication criteria as it currently stands. Therefore, we invite you to submit a revised version of the manuscript that addresses the points raised during the review process.

Authors should follow the recommendations made by the reviewers and myself. Specifically, it is necessary before a recommendation for publication of the manuscript can be made that the authors rewrite their discussion as it is currently primarily an extended results section, not providing an accurate discussion of their results. 

We look forward to receiving your revised manuscript.

Kind regards,

Jose M. Moran

Academic Editor

PLOS ONE

Journal Requirements:

4. Please include additional information regarding the survey or questionnaire used in the study and ensure that you have provided sufficient details that others could replicate the analyses. For instance, if you developed a questionnaire as part of this study and it is not under a copyright more restrictive than CC-BY, please include a copy, in both the original language and English, as Supporting Information.

Furthermore, in your Methods section, please provide a justification for the sample size used in your study, including any relevant power calculations (if applicable).

5. We note that Figure 1 in your submission contain map images which may be copyrighted. All PLOS content is published under the Creative Commons Attribution License (CC BY 4.0), which means that the manuscript, images, and Supporting Information files will be freely available online, and any third party is permitted to access, download, copy, distribute, and use these materials in any way, even commercially, with proper attribution. For these reasons, we cannot publish previously copyrighted maps or satellite images created using proprietary data, such as Google software (Google Maps, Street View, and Earth). For more information, see our copyright guidelines: http://journals.plos.org/plosone/s/licenses-and-copyright.

5.1.    You may seek permission from the original copyright holder of Figure 1 to publish the content specifically under the CC BY 4.0 license. 

5.2.    If you are unable to obtain permission from the original copyright holder to publish these figures under the CC BY 4.0 license or if the copyright holder’s requirements are incompatible with the CC BY 4.0 license, please either i) remove the figure or ii) supply a replacement figure that complies with the CC BY 4.0 license. Please check copyright information on all replacement figures and update the figure caption with source information. If applicable, please specify in the figure caption text when a figure is similar but not identical to the original image and is therefore for illustrative purposes only.

Reviewers' comments:

Reviewer's Responses to Questions

**Comments to the Author**

1. Is the manuscript technically sound, and do the data support the conclusions?

Reviewer #1: Yes

Reviewer #2: Partly

2. Has the statistical analysis been performed appropriately and rigorously? 

Reviewer #1: Yes

Reviewer #2: Yes

3. Have the authors made all data underlying the findings in their manuscript fully available?

Reviewer #1: Yes

Reviewer #2: Yes

4. Is the manuscript presented in an intelligible fashion and written in standard English?

Reviewer #1: Yes

Reviewer #2: No

5. Review Comments to the Author

Reviewer #1: This manuscript presents a very interessant approach about how to evaluate antropometric measures on children and adolescence. The writting is well done , specially on introdution and methods, perphaps some improvements are need on other sections that will be pojnted bellow. In general, the manuscript is eligible to be published, but need some revisions.

Authors states on introduction that they intend to incorporate new antropometric indicators in addition to assessing the nutritional status. It would be very interesting if the could take a more comparative approach of the results found here with the traditional findings in their discussion.

It is of note that they were very cautious on methods since it is it is very well detailed, but please specift why did you decided to do arithmetic average for anatomical point instead of traditional average as you did the other antropometric measures.

The firts paragraph of results (line 237) it is not necessary since the age range of children is very extensive, what can the average add to knowledge? Results b age group and sex is much more plausible. Something similar happens when authors presents the mean wieht comparation between groups by age (line 272) : What is the relevance of information? Children of very different ages will certainly have different weights. Unlike the FMI/ FM comparison that presented great and interesting results that shoul be discussed.

Please, review the discussion thread: do not mention tables and present direct results, use this section to add the new ideas of your study and return to that at the conclusion (the conclusion do not mentioed the use of the new approach on antropometric measures)

FInally, check all commas and points there are serveral errors on discussion text and on table 1 ( mean 2.126 ??)

Reviewer #2: This is an interesting study regarding the anthrometry assessment of body composition and nutritional status in Bangladesh children aged 2-15 years. The sample size is relatively small by considering this study was carried out at 3 districts. This kind of study is important in a developing country to understand growing children's nutritional status and body composition as well as their contributing factors.

Major concerns:

Introduction:

It lengthy, with unnecessary literature and without a main focus. Significance of this study was not pointed out in the introduction. First paragraph can be deleted, and please consolidated these lines into two paragraphs only (line76-106; and line108-143). Please add in the previous findings on associations between anthropometric and nutritional status, identify the gaps in current literature, and what is the significance of this study.

Methods:

Please add in the ethics approval number. More importantly, how do you assess children's nutritional status by anthropometric? What are the cut offs to categorize children into several status of malnutrition?

The cut-off in the discussion (line 384-393): should be included in method section

Do you have dietary and food intake data--it would be interesting and would value up this paper by cross-checking the nutritional status that was determined by anthropometric measurement as well as food intake questionnaire

Results is okay with suitable analysis

Discussion:

The authors need to re-write the discussion. The current version is hard for the reader to follow. The 1st paragraph of the discussion is to demonstrate the main findings of this paper. The authors have a tendency to explain the different anthropometric parameters/ methods use to access children's body composition profiles (e.g. line 394-385, line 360-382; line 396-401; line 409-412) and these are not necessary. Instead, authors should focus on the main findings: (1) whether the anthropometric and nutritional status in Bangladesh children is lower/ higher as compared to the other developing countries with similar SES with reasons and recommendation; (2) the binary logistic results that show us some potential factors that contributing in the poor nutritional status in Bangladesh children, and with some scientific arguments in related to the previous findings. In addition, please don't mention (Table XXX) in discussion--this should only mention in Result section.

Minor concern

Please check your English. A lot of typos

6. PLOS authors have the option to publish the peer review history of their article (what does this mean?). If published, this will include your full peer review and any attached files.

Reviewer #1: No

Reviewer #2: No

---

## [Author Response · Author response to Decision Letter 0]

1 Jul 2021

REBUTTAL LETTER 

16 June 2021

Jose M. Moran

Academic Editor

PLOS ONE

Ref: Manuscript ID: PONE-D-21-12299 

Manuscript Title: The anthropometric assessment of body composition and nutritional status in children aged 2-15 years: A cross-sectional study from three districts in Bangladesh

Dear Jose M. Moran, 

Thank you for your email and for reconsidering our submission. We have addressed the issues raised by the editorial team. A rebuttal letter, which provides our responses, is appended. We hope that the revised manuscript will prove acceptable for publication.

Kind Regards,

Md Kamruzzaman

(On behalf of co-authors) 

POINT-BY-POINT REBUTTAL

Response to Editor

Response: We thank the Associate Editor for this comment. We have checked again the PLOS ONE’S style requirements and ensured the style.

Response: We thank the editor for this comment. We have revised the section related to participant consent and provided more information (Page 8 and Line 318). 

Response: Thanks for this comment. We have copyedited the manuscript language usage, spelling, and grammar. [Edited by Dr Md Rezaul Haque, Professor, Dept. of English, Islamic University, Kushtia, Bangladesh, and Adjunct Associate Professor at St. John's University, New York, USA) 

4. Please include additional information regarding the survey or questionnaire used in the study and ensure that you have provided sufficient details that others could replicate the analyses. For instance, if you developed a questionnaire as part of this study and it is not under a copyright more restrictive than CC-BY, please include a copy, in both the original language and English, as Supporting Information.

Response: We have included the questionnaire in original language (Bengali) and English as supplementary file.

Furthermore, in your Methods section, please provide a justification for the sample size used in your study, including any relevant power calculations (if applicable).

Response: Thanks for this comment. We have now included the details of sample size calculations (Page 7, Line 298)). 

5. We note that Figure 1 in your submission contain map images which may be copyrighted. All PLOS content is published under the Creative Commons Attribution License (CC BY 4.0), which means that the manuscript, images, and Supporting Information files will be freely available online, and any third party is permitted to access, download, copy, distribute, and use these materials in any way, even commercially, with proper attribution. For these reasons, we cannot publish previously copyrighted maps or satellite images created using proprietary data, such as Google software (Google Maps, Street View, and Earth). For more information, see our copyright guidelines: http://journals.plos.org/plosone/s/licenses-and-copyright.

Response: Thanks for this comment. We have now replaced the figure that complies with the CC BY 4.0 license. (Fig 1) (Page 34, Line 1712). 

 

Comments from Reviewer 1 

Reviewer #1: This manuscript presents a very interessant approach about how to evaluate antropometric measures on children and adolescence. The writting is well done , specially on introdution and methods, perphaps some improvements are need on other sections that will be pojnted bellow. In general, the manuscript is eligible to be published, but need some revisions.

Authors states on introduction that they intend to incorporate new antropometric indicators in addition to assessing the nutritional status. It would be very interesting if the could take a more comparative approach of the results found here with the traditional findings in their discussion.

It is of note that they were very cautious on methods since it is it is very well detailed, but please specift why did you decided to do arithmetic average for anatomical point instead of traditional average as you did the other antropometric measures.

The firts paragraph of results (line 237) it is not necessary since the age range of children is very extensive, what can the average add to knowledge? Results b age group and sex is much more plausible. Something similar happens when authors presents the mean wieht comparation between groups by age (line 272) : What is the relevance of information? Children of very different ages will certainly have different weights. Unlike the FMI/ FM comparison that presented great and interesting results that shoul be discussed.

Please, review the discussion thread: do not mention tables and present direct results, use this section to add the new ideas of your study and return to that at the conclusion (the conclusion do not mentioed the use of the new approach on antropometric measures)

FInally, check all commas and points there are serveral

Specific Query and Response

Q1: It would be very interesting if they could take a more comparative approach of the results found here with the traditional findings in their discussion.

Response: We thank the reviewer for their comments. We have revised the discussion section and provided more information (Page 16). 

Q2: It is of note that they were very cautious on methods since it is it is very well detailed, but please specift why did you decided to do arithmetic average for anatomical point instead of traditional average as you did the other antropometric measures.

Response: Thanks for this comment. As far as we know, arithmetic mean and traditional mean are same, and we used this arithmetic mean or traditional mean. However, the little (point) difference that you have noticed is due to the use of different statistical package or function that we have used in RStudio. We hope this little difference won.t be a big problem. 

Q 3

The first paragraph of results (line 237) it is not necessary since the age range of children is very extensive, what can the average add to knowledge? Results b age group and sex is much more plausible. Something similar happens when authors presents the mean wieght comparation between groups by age (line 272) : What is the relevance of information? Children of very different ages will certainly have different weights. Unlike the FMI/ FM comparison that presented great and interesting results that shoul be discussed.:

Response: Thanks for this comment. Average age and weight of children has been deleted from paragraph of result section (Page 11, Line 438).

Q4: Review the discussion thread: do not mention tables and present direct results, use this section to add the new ideas of your study and return to that at the conclusion (the conclusion do not mentioed the use of the new approach on antropometric measures)

Response: Thanks for this comment. The discussion section has been revisited and addressed those issues. (Page 16, Page 23).

Q5 FInally, check all commas and points there are serveral errors on discussion text and on table 1 (mean 2.126 ??)

Response: Thanks for this comment. All error of commas and points has been checked and corrected. (Page 22, Table 1).

Comments from Reviewer 2 

Reviewer #2: This is an interesting study regarding the anthrometry assessment of body composition and nutritional status in Bangladesh children aged 2-15 years. The sample size is relatively small by considering this study was carried out at 3 districts. This kind of study is important in a developing country to understand growing children's nutritional status and body composition as well as their contributing factors.

Major concerns:

Introduction:

It lengthy, with unnecessary literature and without a main focus. Significance of this study was not pointed out in the introduction. First paragraph can be deleted, and please consolidated these lines into two paragraphs only (line76-106; and line108-143). Please add in the previous findings on associations between anthropometric and nutritional status, identify the gaps in current literature, and what is the significance of this study.

Methods:

Please add in the ethics approval number. More importantly, how do you assess children's nutritional status by anthropometric? What are the cut offs to categorize children into several status of malnutrition?

The cut-off in the discussion (line 384-393): should be included in method section

Do you have dietary and food intake data--it would be interesting and would value up this paper by cross-checking the nutritional status that was determined by anthropometric measurement as well as food intake questionnaire

Results is okay with suitable analysis

Discussion:

The authors need to re-write the discussion. The current version is hard for the reader to follow. The 1st paragraph of the discussion is to demonstrate the main findings of this paper. The authors have a tendency to explain the different anthropometric parameters/ methods use to access children's body composition profiles (e.g. line 394-385, line 360-382; line 396-401; line 409-412) and these are not necessary. Instead, authors should focus on the main findings: (1) whether the anthropometric and nutritional status in Bangladesh children is lower/ higher as compared to the other developing countries with similar SES with reasons and recommendation; (2) the binary logistic results that show us some potential factors that contributing in the poor nutritional status in Bangladesh children, and with some scientific arguments in related to the previous findings. In addition, please don't mention (Table XXX) in discussion--this should only mention in Result section.

Minor concern

Please check your English. A lot of typos

Q1: Introduction

It lengthy, with unnecessary literature and without a main focus. Significance of this study was not pointed out in the introduction. First paragraph can be deleted, and please consolidated these lines into two paragraphs only (line76-106; and line108-143).

Response: The introduction section has been revised accordingly (Page 4).

Q2: Please add in the previous findings on associations between anthropometric and nutritional status, identify the gaps in current literature, and what is the significance of this study.

Response: Thanks for this comment. This issue has been addressed (Page 7, Line 278).

Q3 Methods:

Please add in the ethics approval number. More importantly, how do you assess children's nutritional status by anthropometric? What are the cut offs to categorize children into several status of malnutrition?

The cut-off in the discussion (line 384-393): should be included in method section

Response: Thanks for this comment. Ethical approval no has been added to the method section (Page 8, Line 327). Details of anthropometric cut-off value has been added in details in anthropometric measurement of the method section (Page 9, Line 358).

Q4: Do you have dietary and food intake data--it would be interesting and would value up this paper by cross-checking the nutritional status that was determined by anthropometric measurement as well as food intake questionnaire

Response: Thanks for raising this nice issue. Indeed, it was interesting and valuable to include dietary and food intake data. However, we didn’t collect those kinds of data. This is beyond are study objectives. We will try to include this kind of date for cross validation of nutritional status of children in future study. 

Q5: Results is okay with suitable analysis.

Response: Thanks for this comment.

Q6: Discussion:

The authors need to re-write the discussion. The current version is hard for the reader to follow. The 1st paragraph of the discussion is to demonstrate the main findings of this paper. The authors have a tendency to explain the different anthropometric parameters/methods use to access children's body composition profiles (e.g. line 394-385, line 360-382; line 396-401; line 409-412) and these are not necessary.

Response: Thanks for this comment. The discussion has been rewritten and checked for error.

Q7: Instead, authors should focus on the main findings: (1) whether the anthropometric and nutritional status in Bangladesh children is lower/ higher as compared to the other developing countries with similar SES with reasons and recommendation.

Response: Thanks for this comment. The issue has been addressed and discussed in detail in the discussion section. (Page 16, 17, 19, 20, 21) 

Q8: (2) the binary logistic results that show us some potential factors that contributing in the poor nutritional status in Bangladesh children, and with some scientific arguments in related to the previous findings.

Response: Thanks for this comment. The issue has been addressed and discussed in detail in the discussion section. (Page 21, Line 1124) 

Q9: Minor concern, please check your English. A lot of typos

Response: The manuscript has been rechecked for language and typos.

---

## [Decision Letter · Decision Letter 1]

26 Jul 2021

PONE-D-21-12299R1

The anthropometric assessment of body composition and nutritional status in children aged (2-15 years): A cross-sectional study from three districts in Bangladesh

PLOS ONE

Dear Dr. Kamruzzaman,

Thank you for submitting your manuscript to PLOS ONE. After careful consideration, we feel that it has merit but does not fully meet PLOS ONE’s publication criteria as it currently stands. Therefore, we invite you to submit a revised version of the manuscript that addresses the points raised during the review process.

There are minor issues that need to be addressed before the manuscript can be recommended for publication. Those detailed by reviewer #2 will provide clarity and a better understanding of the manuscript.

We look forward to receiving your revised manuscript.

Kind regards,

Jose M. Moran

Academic Editor

PLOS ONE

Journal Requirements:

Reviewers' comments:

Reviewer's Responses to Questions

**Comments to the Author**

1. If the authors have adequately addressed your comments raised in a previous round of review and you feel that this manuscript is now acceptable for publication, you may indicate that here to bypass the “Comments to the Author” section, enter your conflict of interest statement in the “Confidential to Editor” section, and submit your "Accept" recommendation.

Reviewer #1: All comments have been addressed

Reviewer #2: All comments have been addressed

2. Is the manuscript technically sound, and do the data support the conclusions?

Reviewer #1: Yes

Reviewer #2: Yes

3. Has the statistical analysis been performed appropriately and rigorously? 

Reviewer #1: Yes

Reviewer #2: Yes

4. Have the authors made all data underlying the findings in their manuscript fully available?

Reviewer #1: Yes

Reviewer #2: Yes

5. Is the manuscript presented in an intelligible fashion and written in standard English?

Reviewer #1: Yes

Reviewer #2: Yes

6. Review Comments to the Author

Reviewer #1: The manuscript was properly reformulated, the language was corrected, the reviewers' questions were answered. The current version is subject to publication.

Reviewer #2: I think this manuscript is ready for publication. The authors addressed the comments from both reviewers and editor.

However, there are some minor concerns.

Introduction: Can authors shorten and re-write the 1st two paragraphs in Introduction section? They are lengthy and without a specific focus. 1st paragraph (from line 87-118): should focus on the malnutrition in children at developing countries instead of worldwide, while 2nd paragraph (from line 120-153): please put the focus on the anthropometric assessments in children, as well as the one that you used in the current study, rather than a literature review on all the available methods in human.

Discussion:

1st paragraph (line 388-397) I would think line 388-394 are not necessary, suggest to remove. Please keep line 394-397 and please summarise the main findings of this study in your 1st paragraph of your discussion section.

Line 399-410, can you summarise the main message from these lines. This is too lengthy

Line 413-415: Please check the sentence. Wrong placement for punctuation marks and capital letters.

7. PLOS authors have the option to publish the peer review history of their article (what does this mean?). If published, this will include your full peer review and any attached files.

Reviewer #1: No

Reviewer #2: No

---

## [Author Response · Author response to Decision Letter 1]

28 Jul 2021

REBUTTAL LETTER 

27 July 2021

Jose M. Moran

Academic Editor

PLOS ONE

Ref: Manuscript ID: PONE-D-21-12299R1 

Manuscript Title: The anthropometric assessment of body composition and nutritional status in children aged 2-15 years: A cross-sectional study from three districts in Bangladesh

Dear Jose M. Moran, 

Thank you for your email and for reconsidering our submission. We have addressed the issues raised by the editorial team. A rebuttal letter, which provides our responses, is appended. We hope that the revised manuscript will prove acceptable for publication.

Kind Regards,

Md Kamruzzaman

(On behalf of co-authors) 

POINT-BY-POINT REBUTTAL

Response to Editor

Response: We thank the Associate Editor for this comment. The reference has been thoroughly checked again. No retracted article was found among the reference lists.

 

Comments from Reviewer 2 

Reviewer #2: 

Q1: Introduction

Introduction: Can authors shorten and re-write the 1st two paragraphs in Introduction section? They are lengthy and without a specific focus. 1st paragraph (from line 87-118): should focus on the malnutrition in children at developing countries instead of worldwide, while 2nd paragraph (from line 120-153): please put the focus on the anthropometric assessments in children, as well as the one that you used in the current study, rather than a literature review on all the available methods in human.

Response: The introduction section has been revised accordingly (Page 4, Line 99).

Q2: Discussion:

1st paragraph (line 388-397) I would think line 388-394 are not necessary, suggest removing. Please keep line 394-397 and please summarise the main findings of this study in your 1st paragraph of your discussion section.

Response: Thanks for this comment. The discussion section has been revised and shorten accordingly Line 388-397 has been removed and main findings of this study has been summarized. (Page 16, Line 373).

Q3: Line 399-410, can you summarise the main message from these lines. This is too lengthy

Response: The section has been revised and summarized accordingly (Page 16, Line 386).

Q4: Line 413-415: Please check the sentence. Wrong placement for punctuation marks and capital letters.

Response: The section has been revised accordingly (Page 16, Line 390).

---

## [Editor Report · Decision Letter 2]

23 Aug 2021

The anthropometric assessment of body composition and nutritional status in children aged 2-15 years: A cross-sectional study from three districts in Bangladesh

PONE-D-21-12299R2

Dear Dr. Kamruzzaman,

We’re pleased to inform you that your manuscript has been judged scientifically suitable for publication and will be formally accepted for publication once it meets all outstanding technical requirements.

Kind regards,

Jose M. Moran

Academic Editor

PLOS ONE
---

## [Editor Report · Acceptance letter]

25 Aug 2021

PONE-D-21-12299R2 

The anthropometric assessment of body composition and nutritional status in children aged 2-15 years: A cross-sectional study from three districts in Bangladesh 

Dear Dr. Kamruzzaman:

I'm pleased to inform you that your manuscript has been deemed suitable for publication in PLOS ONE. Congratulations! Your manuscript is now with our production department. 

Kind regards, 

on behalf of

Dr. Jose M. Moran 

Academic Editor

PLOS ONE